# Improved Rotational Foam Molding Properties of Tailored Polyethylene Blends with Higher Crystallization Temperature and Viscosity-Temperature Sensitivity

**DOI:** 10.3390/polym14173486

**Published:** 2022-08-25

**Authors:** Xuelian Chen, Qigu Huang

**Affiliations:** State Key Laboratory of Chemical Resource Engineering, Key Laboratory of Carbon Fiber and Functional Polymers, Ministry of Education, The College of Material Science and Engineering, Beijing University of Chemical Technology, Beijing 100029, China

**Keywords:** polyethylene, rotational molding, foams, crystallization, viscosity–temperature sensitivity, morphology, cell sizes

## Abstract

Rotational foam molding has attracted more and more attention due to the inexpensive machines required, low residual stresses, and flexible design for special and high-value-added applications. However, it is a great challenge to control cell sizes and morphology because of its coalesce and collapse during prolonged heating or at different temperatures. A novel tailored polyethylene blend with a unique chain structure for rotational foam molding was creatively proposed and demonstrated, and the effects of crystallization temperature and viscosity–temperature sensitivity on foaming were also investigated. The polyethylene blends with few chain branches in the low-molecular-weight part and many chain branches in the high-molecular-weight part effectively improved the crystallization temperature and the viscosity–temperature sensitivity for better prevention of coalesce and collapse during the foam-shaping stage.

## 1. Introduction

Recently, polymeric foams have attracted extensive attention in the industry due to their numerous benefits, such as low cost per unit volume, superior thermal insulation, low coefficient of thermal expansion, excellent sound isolation, and high stiffness for a given weight. [1,2]. The global market of polyethylene foams is projected to grow rapidly from USD 3.6 billion in 2021 to reach USD 4.8 billion by 2026 [1]. They are widely used in automotive, construction, logistics packaging, sports equipment, medical equipment, 5G, and other fields [3,4,5,6]. To meet more and more special requirements, such as use furniture, containers, fuel tanks, toys, and kayaks, many large and hollow seamless foamed articles are produced via rotational molding. Rotational molding is a pressureless process used to produce hollow plastic articles with a complex structure. The main advantages of rotationally molded articles are a lack of weld lines, low residual stresses, and flexible design with a wide range of multi-layers, sizes, thicknesses, and shapes. Compared with other processes, including injection molding, blow molding, and extrusion, multi-layer end-used articles can be produced with inexpensive equipment by rotational foam molding [7,8,9]. However, there are also some disadvantages, including a long processing time, limited processing pressure, and few suitable materials [6,7,8,9,10,11,12,13,14,15,16]. Although the rotational-molding technique has been developed for more than five decades, research efforts in rotational foam molding are still limited.

The mechanism of the foaming process is the essential universal prerequisite for fabricating high-quality foamed plastic articles [11,12,13,14,15]. The general foaming processes include nucleation, growing, solidifying, and shaping. The blowing agent decomposes and generates gas in the polyethylene matrix to form cell nucleation with increasing temperature, and then these gas bubbles continue to grow with little resistance. Neighboring bubbles coalesce as the chemical blowing agent decomposes and the void fraction increases. The nucleation process includes two stages, primary and secondary nucleation, and the nucleation rate is influenced by the melt viscosity of the polymer materials [11]. A lower melt viscosity of the polymer matrix could allow a greater number of bubbles to survive and grow larger than the critical nucleus, and higher melt viscosity could impede the number of nuclei generated. For the bubble lifespan and bubble diameter, a higher temperature reduces the bubble lifespan and generates smaller bubble diameters, whereas a higher chemical blowing agent’s concentration results in a longer bubble lifespan and a larger bubble diameter [12]. In addition, the melt viscosity in a certain range has little effect on the bubble lifespan but inhibits the bubble growth and the collapse rate, which seems beneficial for foaming [13].

There are several challenges to producing fine, uniform cells and high-expansion-ratio foams for the rotational-molding process. Firstly, it is well-known that only atmospheric pressure is applied during the conventional rotational-molding process. This will lead to relatively few bubble nucleations, the dimensions of bubble nucleation are typically greater than 150 μm, and the bubble growth only depends on the gas diffusion. The power of gas diffusion mainly comes from the competition between the internal pressure produced by the decomposition gas of the foaming agent and the external pressure of the polymer melt. Secondly, the temperature in the polymer melt is not uneven and changes significantly with the increase in processing time. Thirdly, the presence of cells in the foamed products results in a reduced heat-conduction velocity and a longer cooling process. Too long of a cooling time will cause coalesce and collapse in the polymer melt and make the surface of the foamed article less smooth.

The melt characteristics of polymer play important roles in determining the foaming properties. Lower shear viscosity can facilitate sintering behavior. However, very low shear viscosity may not be desirable for the foaming process because the associated reduction in melt strength will lead to severe deformed cellular structure, so it is very difficult to select suitable polyethylene resins for pressureless foaming [16,17]. Metallocene-catalyzed polyethylene is less sensitive to the demoulding temperature than Ziegler–Natta-catalyzed polyethylene, which is beneficial for producing foams with more uniform cell size and provides a much wider processing window [16,17,18]. Nucleation and growth of bubbles should occur as simultaneously as possible to keep the bubble size almost the same and reduce cell coarsening [19]. Most HDPE foams produced by rotational molding with different molecular weights had closed-cell structures. The foaming grade was decreased with the increase in molecular weight, and cell sizes were also highly dependent on the molecular structure and the concentration of chemical blowing agents [20].

To improve the pressureless foaming properties of high-melt-index polyethylene, different methods have been proposed to avoid cell coalescence by increasing the melt viscosity [21,22,23]. In fact, chemical-crosslinking technology is widely used to control the pressureless-foaming process to enhance the melt strength and elongation viscosity. The crosslinked foams exhibit high expansion ratios and narrow cell size distributions. Enhanced strain-hardening behavior via crosslinking in the melt viscosity prevents local cell deformation. However, it is very difficult to control the foaming process via chemical crosslinking in an industrial setting because both crosslinking and foaming are dependent on temperature and polyethylene formulation simultaneously. Excessive crosslinking restricts foam expansion, while insufficient crosslinking results in bubble rupture. The effect of the molecular architecture on the cellular structure of crosslinked polyethylene was investigated by Abe and Yamaguchi [21]. It was found that the expansion ratio of the crosslinked LLDPE foam is decided by the elastic modulus and the crystallization temperature. The degree of shrinkage decreases when increasing the crystallization temperature because immediate crystallization prevents shrinkage. 

However, to the best of our knowledge, no studies reported the relationship between molecular structure, crystallization temperature, viscosity–temperature sensitivity, and rotational-foaming properties of the polyethylene blends. The main objective of this work is to design and produce novel, tailored polyethylene blends with unique structures for rotational foam molding. In particular, the effects of the polymeric structure on crystallization temperature and viscosity–temperature sensitivity will be investigated, as well as their effects on foaming properties, including cell size and morphology.

## 2. Materials and Methods

### 2.1. Materials and Methods

The raw polyethylene materials (M20/D954, M20/D924, M2/D918, M8/D965) were supplied by Shenhua Coal Chemical Industrial Company, the raw material labeled M0.5/D868 was purchased from Dow Chemical Company, and the polyethylene labeled M0.06/D948 was purchased from Saudi Polymers. Herein, M and D represent the melt index and density of raw materials in Table 1, which are measured according to ASTM D1238 and ASTM D 1505, respectively. Six polyethylene blends from PE-B1 to PE-B6 were prepared for this study. They were produced by melt-blending with different raw polyethylene materials, as described above. These samples were designed to represent different rheological properties and crystallization temperatures due to their different polymeric structures. Their formulas are shown in Table 1, and their basic properties are described in Table 2.

The foaming agent was a commercialized azodicarbonamide (AZ), provided by Adamas Reagent Co., Ltd., Shanghai, China. The particle size of the industrial grade ranged from 12 μm to 18 μm, and the average particle size was about 13.25 μm. The gas yield was 220 mL/g, and its decomposition temperature was closed to 205 °C. Zinc oxide (ZnO) powder, provided by Sinopharm Chemical Reagent Co., Ltd., Shanghai, China, was selected as an activator to modify the decomposition temperature of the blowing agent for this foaming system. Based on the resin mass of 100 phr, the amounts of AZ and zinc oxide were 3 phr and 0.12 phr, respectively. The foaming agents and polyethylene blends were mixed under the setting temperature of 135 °C using a twin-screw extruder. The extruder has a diameter of 26 mm and an L/R ratio of 40. Then, the polyethylene blends with the additives were milled to 35 mesh under 60 °C for rotational foam-molding.

### 2.2. Rotational Foam-Molding Procedure

Rotational foam-molding experiments were carried out on a laboratory-scale bi-axially rotating machine with an electrically heated oven and a circulation fan. The diameter of the oven was 1500 mm, and a 500 mm × 500 mm × 300 mm box mold made of a thick sheet of aluminum was used for all foam-molding tests. The whole process involved the following: The polyethylene samples were placed into the mold, and we put the mold into the oven at a suitable temperature for heating for the required time. Then, we took the mold out of the oven to cool using the circulation fan and removed the rotomolded article from the mold. The oven temperatures could reach 300 °C. Herein, the oven temperature was under 285 °C, and the heating time was 25 min. Any specimens were cut out of these articles.

### 2.3. Testing Procedure

Gel permeation chromatography (GPC) was carried out using a Polymer Char machine with a multi-detector to measure the molecular weight (Mw) and molecular weight distribution (MWD). The measurement was carried out at 150 °C at a flowing rate of 1.0 mL/min. In GPC analysis, the number-average molecular weight (Mn), weight-average molecular weight (Mw), and z-average molecular weight (Mz) can be simultaneously determined. Furthermore, the Mw/Mn ratio is used to evaluate the molecular weight distribution (MWD). All GPC curves are showed in Figure 1, and the values of molecular weight are listed in Table 2.

The polymer crystallization elution fraction (CEF) test was carried out using a Polymer Char CEF machine with a viscosity detector and an IR6 detector. A total of 8 mg of polyethylene sample was dissolved into 10 mL1,2,4-trichlorobenzene at160 °C for 180 min, and then 200 μL of the solution was injected into the instrument at 150 °C, rapidly cooled to 95 °C, and then procedurally cooled from 95 °C to 35 °C at a rate of 2 °C/min, and the flow rate was kept at 0.05 mL/min. The heating process was from 35 °C to 150 °C at a rate of 4 °C/min, and the flow rate of the heating section was 1.0 mL/min. All CEF curves are showed in Figure 2.

Rheological measurements were performed using a controlled strain rheometer (TA instruments) in oscillatory mode with a parallel plat fixture (20 mm in diameter) at a gap of 1.0 mm under a nitrogen atmosphere. The frequency sweep model was used at 200 °C over an angular frequency range of 0.05–100 rad/s, and the viscosity-frequency curves are showed in Figure 3a. The temperature sweep model was used from 150 °C to 200 °C at a heating rate of 10 °C/min under a strain of 5% and a frequency of 1.0 Hz, and the viscosity-temperature curves are showed in Figure 3b. For the rheological measurements, the samples were prepared using a hot press (P300P series, Collin Inc., USA) at 160 °C and 12 MPa for 5 min and then cooled to 30 °C at a rate of 10 °C/min.

Viscous flow activation energy (Ea) is used to describe the viscosity–temperature dependence of a polymer using the rheometer. Its apparent viscous flow activation energy is higher, so the influence of temperature on the characteristics of the melt is stronger. 

The shear-viscosity data were obtained by melt rheology experiments. The relationship between the apparent shear viscosity of the polymer and the temperature is in accordance with the Arrhenius empirical formula. The curves of the melt shear viscosity of the six polyethylene blends with temperature are shown in Figure 3b, and the value of Ea can be easily calculated in Table 3.

Differential scanning calorimetry (DSC) analysis was carried out using a Q2000 instrument to evaluate the thermal properties of the polyethylene blends before the foaming process. Specimens at least 6 mg in weight were placed in aluminum pans. The heating and cooling runs were performed from 20 °C to 200 °C under a nitrogen atmosphere at a heating and cooling rate of 10 °C/min. All DSC crystallization curves are showed in Figure 4.

The foam expansion ratio was calculated based on the following equation: VER = ρ/ρf. Herein, ρ and ρf represent the density of polyethylene blends before and after foaming. Density was measured by weighing each sample with a precision of 0.0001 mg and calculating its volume as the volume of water displaced in a calibrated pycnometer where the sample was submerged.

To determine the cell size (CS) and cell diameter distribution (CSD), a micrograph of each sample was taken using an optical microscope. The cell diameter distribution (CSD) was also obtained by measuring and counting the observed bubbles. All cells morphology of foamed parts is showed in Figure 5.

## 3. Results

### 3.1. Concept Validation

Two kinds of PE blends with different structures were designed and prepared in the first round of trials to validate our hypothesis. One was PE-B2, with few chain branches in the high-molecular-weight part and many chain branches in the low-molecular-weight part. Contrary to the polymeric chain structure of PE-B2, the other sample labeled PE-B4 had a unique chain structure with many chain branches in the high-molecular-weight part and few chain branches in the low-molecular-weight part. 

The molecular weight (M_W_) of PE-B2 was higher than that of PE-B4, which was consistent with the value of the melt index (MI_2.16_), and the molecular weight distribution (MWD) of PE-B2 was wider than that of PE-B4, which was also consistent with the value of MI_21.6_/MI_2.16_ in Table 2 and Figure 1a. The results indicated that the average molecular chain length of PE-B2 was longer than that of PE-B4, but the molecular chain distribution of PE-B2 was more uneven. The CEF results are shown in Figure 2a, and they indicated that the molecular chain structures of PE-B2 and PE-B4 were quite different. The eluted fractions under 90–100 °C of PE-B4 were much higher than that of PE-B2, but the eluted fractions under 70–80 °C and soluble fractions were significantly less than those of PE-B2. This result indicates that more branched-chain components were in PE-B2, while PE-B4 contained fewer branched chains.

The rheological properties of PE-B2 and PE-B4 are shown in Figure 3. The complex viscosity of PE-B2 was higher than that of PE-B4, which implied that PE-B2 had higher melt strength in melting viscosity due to its higher molecular weight. It was commonly believed that the higher complex viscosity (or elastic modulus) implied a stronger ability of the polymer to support better foamability [24]. However, a higher melt strength could not support good morphology for PE-B2. Although the viscosity of PE-B2 was higher, the morphology of cells was not well-controlled. Figure 5 shows that the cell size was large, and the collapse was serious. This phenomenon was attributed to the viscous flow activation energy (Ea) and crystallization temperature. PE-B4 showed much higher viscous flow activation energy (Ea) in Table 3 than did PE-B2. Viscous flow activation energy (Ea) is used to describe the viscosity–temperature dependence of a polymer. Further implied is that the melt viscosity of PE-B4 decreased exponentially with the increase in temperature, and the gas-diffusion speed decreased with the increase in viscosity during the melt’s cooling. The crystallization curves of PE-B2 and PE-B4 are given in Figure 4a. The crystallization temperature of PE-B2 was lower than that of PE-B4, and the crystallization rate of PE-B2 was also lower than that of PE-B4. The high crystallization temperature and crystallization rate of the polymer melt during the shaping of foam cells will significantly enhance the melt strength around the foam cells to avoid fusion and collapse to make the foam cells more uniform.

Although the viscosity of PE-B2 was higher, the morphology of cells was not well-controlled, the cell size was large, and the collapse was serious (Figure 5). The results showed that the expand ratios of PE-B2 and PE-B4 were about 8, but the average cell sizes of PE-B2 and PE-B4 were significantly different. The average cell size of PE-B2 was about 1000 μm, while that of PE-B4 was only 570 μm. This result can be due to their different structures. There was a high branching degree of low-molecular-weight components and low branching degree of high-molecular-weight components in PE-B2, while PE-B4 had a low branching degree of low-molecular-weight components and a high branching degree of high-molecular-weight components. The molecular structure of PE-B4 can provide higher viscosity–temperature sensitivity and higher crystallization temperature, which are beneficial for cell shaping during rotational foam-molding.

### 3.2. Effects of Polymer Components

Polyethylene with few chain branches in the low-molecular-weight part and many chain branches in the high-molecular-weight part was better at controlling the cell size and morphology for rotational foam molding. However, it is a challenge to choose the polymer components. The effects of polymer components on viscosity–temperature sensitivity, crystallization behavior, and foaming properties were further investigated. Different polyethylene resins were selected to prepare the blends with few chain branches in the low-molecular-weight part and many chain branches in the high-molecular-weight part. Three samples, PE-B1, PE-B3, and PE-B4, were designed and prepared for this study. Compared with PE-B4, PE-B1 was chosen for its lower density and lower melt index resin to provide a high-molecular-weight component with a high number of branched chains, and PE-B3 was used for its higher density and lower melt index resin, which provided a low-molecular-weight component with a lower number of branching-degree chains. 

From the analysis of the molecular weight and molecular weight distribution in Table 2 and Figure 1a, the molecular weight of PE-B3 was slightly greater than PE-B1 and significantly greater than PE-B4. There was no significant difference in the molecular weight distribution. The CEF data in Figure 2a show that the eluted fraction of PE-B1 is the same as that of PE-B4 under 90–100 °C, while PE-B1 showed a greater eluted fraction than PE-B4 under lower temperatures. These data indicate that PE-B1 contained many more non-crystalline components. 

PE-B3 has a significantly higher melt viscosity than PE-B4, which is consistent with the molecular weight and the melting index, as shown in Figure 3a. However, PE-B4 provided higher viscous flow activation energy (Ea) than PE-B1 or PE-B3, as listed in Table 3 and Figure 3b. The three samples were all extremely similar in their crystallization temperature and crystallization rate, except for only a slight difference in crystallinity, as shown in Figure 4a. This indicated that the three samples would form round cells in the foamed articles in Figure 5. Due to the addition of some higher-molecular-weight elastomer into PE-B1 blends, the melt viscosity was too high to inhibit the gas diffusion and reduce the expansion ratio to about 4. The other two samples of PE-B3 and PE-B4 had a good foaming performance with an expansion ratio of about 8. Their cell sizes were 710 μm and 570 μm, respectively, and the cell size distributions were very narrow. 

### 3.3. Effects of Component Proportions

The effects of component proportions on the viscosity–temperature sensitivity, crystallization behavior, and foaming properties were also investigated herein. There were two components in the novel polyethylene blends; one was a low-molecular-weight component with a low branching degree, and the other one was a high-molecular-weight component with a high branching degree. The melt index, molecular weight, and branching degree increased significantly with the increase in the ratio of the high-molecular-weight component. Meanwhile, the eluted fraction near 90–100 °C gradually decreased, along with the eluted fraction near 70–80 °C, and the soluble fraction increased. The melt viscosity increased significantly with the proportion of the high-molecular-weight component; the Ea slightly decreased, as listed in Table 3. The crystallization temperature and crystallization rate also decreased, as shown in Figure 3b. 

In terms of the foaming properties, the three samples (PE-B4, PE-B5, PE-B6) prepared foamed articles with an expansion ratio of about 8–9. However, there were some differences in the morphology and cell sizes. With the increase in the high-molecular-weight component with a high branching degree, the bubbles collapsed, and the morphology became worse, as shown in Figure 5. The crystallization temperature and crystallization rate of polyethylene blends were significantly decreased with the increasing proportion of the high-molecular-weight component. This caused the melt strength to be inadequate around bubbles during the late growth and foam-shaping stage of foaming, so it was difficult to effectively control the morphology of foam cells.

## 4. Conclusions

In this paper, a novel method to control morphology and cell sizes during the foam-cooling stage was revealed, and tailored polyethylene blends with unique structures were proposed and prepared. Novel tailored polyethylene blends with few chain branches in the low-molecular-weight part and many chain branches in the high-molecular-weight part effectively improved the crystallization temperature and the viscosity–temperature sensitivity. Higher crystallization temperature and higher viscosity–temperature sensitivity effectively increased the melt strength around bubble cells during the foam-shaping stage to better prevent coalescence and collapse. The experimental polyethylene blends with a higher melt index (MI_2.16_ > 8 g/10 min) first produced a larger expansion ratio, around 8, with smaller cell sizes (<600 μm) and a narrow cell size distribution. A higher melt index is very important for rotational foam molding when fabricating complex-structure products. However, the relationship between the composition and viscosity–temperature sensitivity needs further investigation.

## Figures and Tables

**Figure 1 polymers-14-03486-f001:**
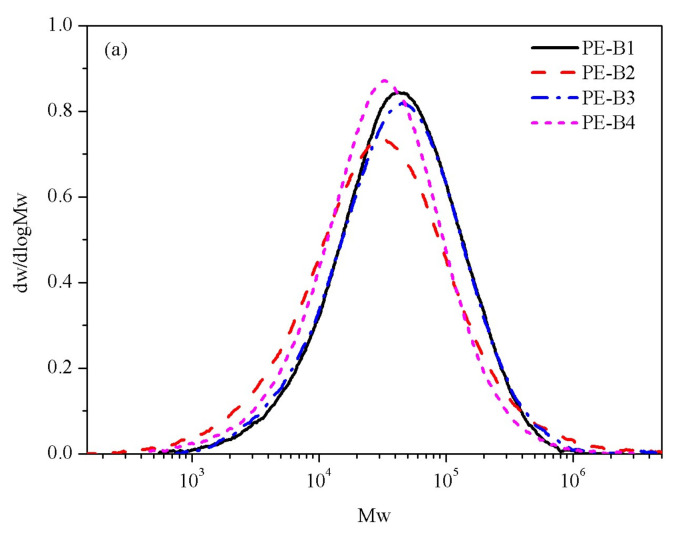
Molecular weight and Molecular weight distribution of samples (**a**) PE-B1, PE-B2, PE-B3, PE-B4; (**b**) PE-B4, PE-B5, PE-B6.

**Figure 2 polymers-14-03486-f002:**
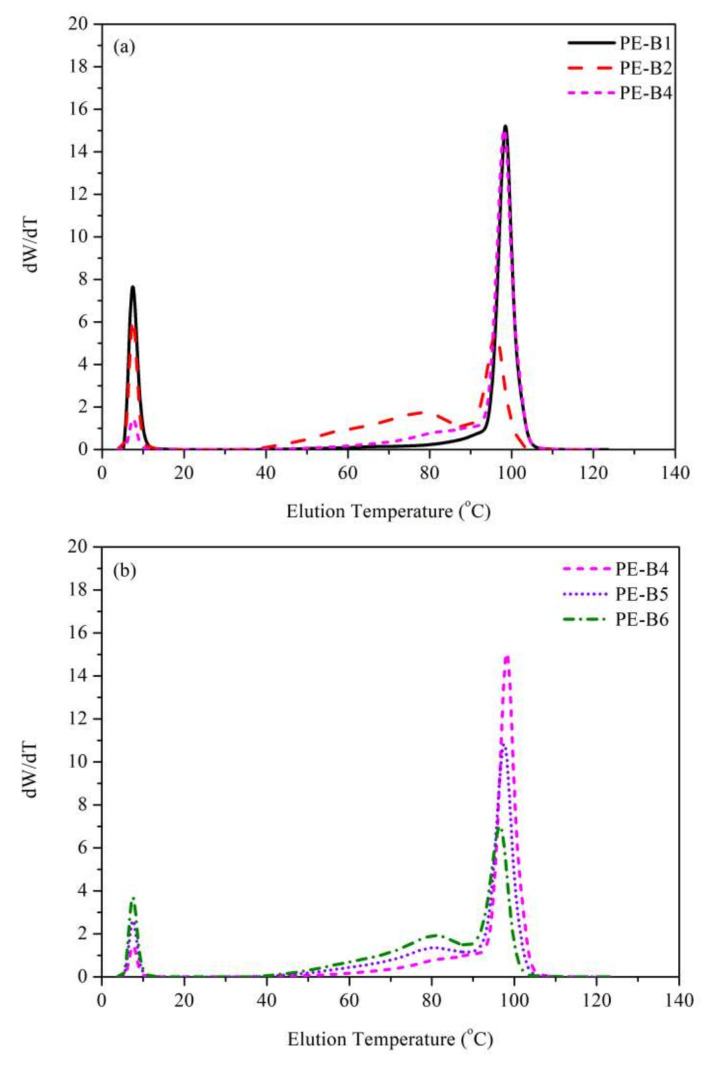
CEF characterizations of samples (**a**) PE-B1, PE-B2, PE-B4; (**b**) PE-B4, PE-B5, PE-B6.

**Figure 3 polymers-14-03486-f003:**
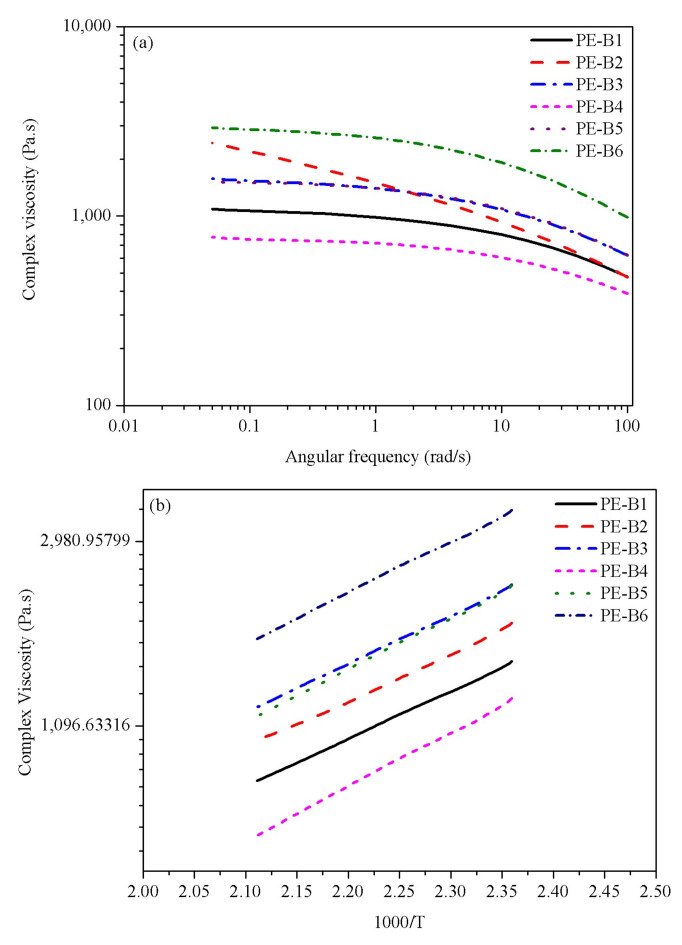
(**a**) Viscosity-frequency curves of samples; (**b**) Viscosity-temperature curves of samples.

**Figure 4 polymers-14-03486-f004:**
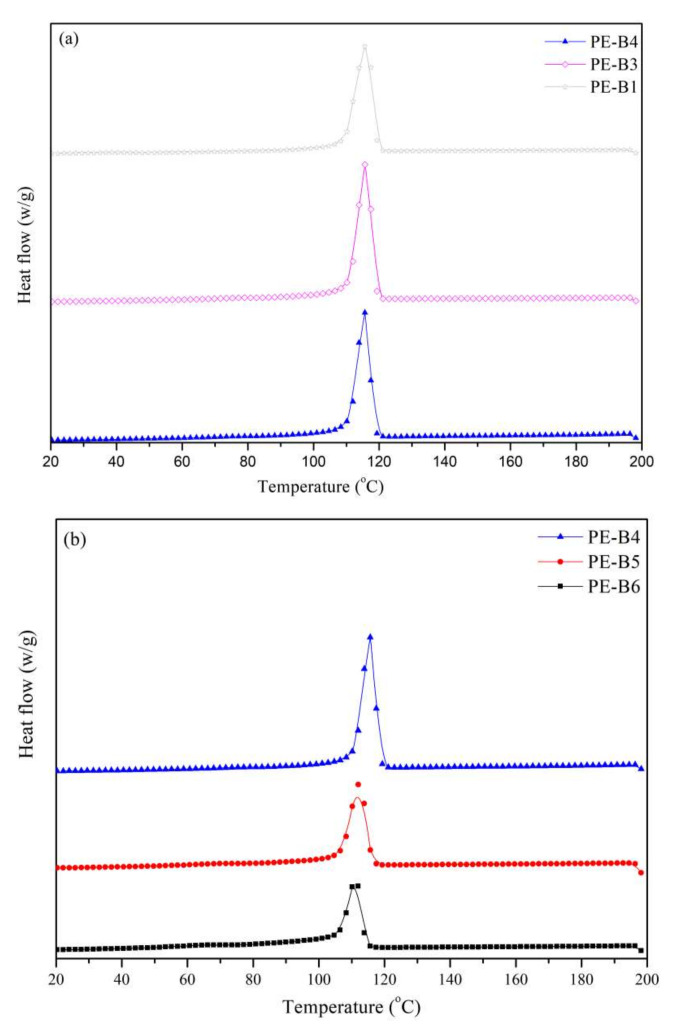
DSC Crystallization curves of samples (**a**) PE-B1, PE-B3, PE-B4; (**b**) PE-B4, PE-B5, PE-B6.

**Figure 5 polymers-14-03486-f005:**
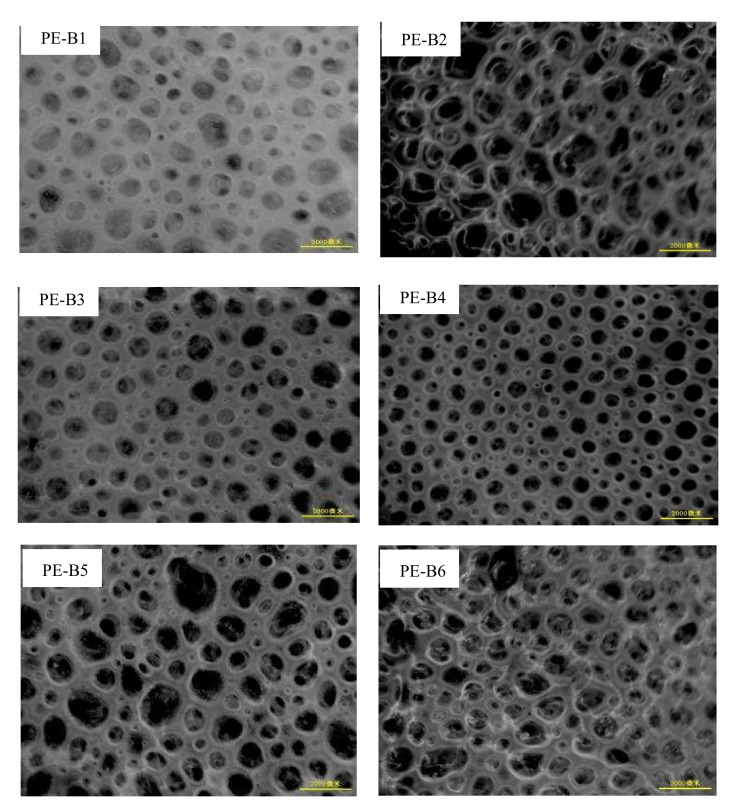
Cells morphology of foamed parts.

**Table 1 polymers-14-03486-t001:** Basic formula of polyethylene blends.

	M20/D954	M20/D924	M2/D918	M0.06/D948	M0.5/D868	M8/D965
PE-B1	80%				20%	
PE-B2		80%		20%		
PE-B3			20%			80%
PE-B4	80%		20%			
PE-B5	50%		50%			
PE-B6	20%		80%			

**Table 2 polymers-14-03486-t002:** Physical and foaming properties of polyethylene blends.

	MI_2.16_	MI_21.6_/MI_2.16_	Mn	Mw	MWD	IV	Rg
PE-B1	11.2	22.1	20837	72561	3.5	1.52	11.7
PE-B2	8.3	24.9	11547	92328	8.0	1.86	11.9
PE-B3	8.1	20.5	20588	78724	3.8	1.50	11.9
PE-B4	15.8	18.3	15144	59860	4.0	1.47	10.5
PE-B5	5.3	26.6	15878	64743	4.1	1.90	11.6
PE-B6	4.1	18.0	20883	91754	4.4	1.94	13.6

**Table 3 polymers-14-03486-t003:** Rheological and crystallization properties of polyethylene blends.

	Ea/KJ·mol^−1^	Tc, Peak/°C	Tc, on Setting/°C	∆ Hc/Jg^−1^
PE-B1	19.94	117.2	119.47	170.79
PE-B2	18.01	113.0	116.31	97.03
PE-B3	21.03	117.1	119.17	182.58
PE-B4	34.56	116.4	118.75	164.12
PE-B5	31.81	113.0	116.18	122.8
PE-B6	31.58	111.7	114.33	105.99

## Data Availability

The data presented in this study are available on request from the corresponding author.

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
