# Peer review of "Improved Rotational Foam Molding Properties of Tailored Polyethylene Blends with Higher Crystallization Temperature and Viscosity-Temperature Sensitivity"

_polymers, 2022, doi:10.3390/polym14173486_

Round 1

Reviewer 1 Report

The current work on  rotational foam molding properties of polyethylene blends is an interesting work on the improving the foamability of PE in a pressureless process. However some modification is needed before publication.

1.      The structures of polymers are vague.

2.      Page 3, L121:" Various additives and resins were mixed under 180oC using with a twin-screw extruder" . There is possibility to some degradation of AZO due to presence of activator. .

3.      Page 5, L202:  How you measure the viscous flow activation energy? Why the authors though this parameter affected on cell size ? Please describe that when cell nucleation and growth occurred? I thought they would conduct during heating, not cooling, and in a semi-isotherm condition. So, I think the justification of the structure with the viscous flow activation energy is not correct.

4.      The cell sizes and expansion ratios of samples should be reported accurately.

5.      The scale bar of SEM images are not clear.

Author Response

Thanks for your kind suggestions. I give my points one by one.

  1. The structures of polymers are vague.

In fact, the novel structure of polymers is a tailored multi-component blends, and the tailored multi-component blends with few chain branches in low molecular weight part and many chain branches in high molecular weight part. We design this unique structure and adjust the ratio between different components to control the crystallization behavior and the viscosity-temperature sensitivity for better preventing the coalesce and collapse during foam-shaping stage.          

  1. Page 3, L121:" Various additives and resins were mixed under 180oC using with a twin-screw extruder" . There is possibility to some degradation of AZO due to presence of activator.

This is a silly mistake. The extruding temperature is 135 oC, not 180 oC.    

3.1 How you measure the viscous flow activation energy?

The shear viscosity data were obtained by melt rheology experiments. The relationship between the apparent shear viscosity of the polymer and the temperature is in accordance with the Arrhenius empirical formula:

Where  is apparent shear viscosity, Pa.s. A is a constant, is the viscous flow activation energy, kj/mol. R is gas constant, and 8.314 J/(mol.K). T is absolute temperature, K.

The curves of the melt shear viscosity of the six polyethylene blends with temperature were showed in Fig.3(b), and  can be easily calculated in Table 3.

3.2 Why the authors though this parameter affected on cell size ? 

We use this parameter to affect the morphology, not only cell size. Higher viscous flow activation energy, indicates higher viscosity-temperature sensitivity. During the cooling process or the foam-shaping stage, if the temperature slightly decreased, the melt viscosity (melt strength) increased significantly,which is benefit for controlling the collapse and preventing the bubble coalesce.        

3.3 Please describe that when cell nucleation and growth occurred? I thought they would conduct during heating, not cooling, and in a semi-isotherm condition.

Yes, I agree with you on that the cell nucleation and growth occurred during heating. This is a common sense and investigated in the previous literature.However, to the best of our knowledge, few studies reported the cooling stage on the cell size and morphology. If the polymer provide slower crystallization and lower viscosity-temperature sensitivity , the cells will collapse or coalesce. In this papers, we proposed a unique structure with few chain branches in low molecular weight part and many chain branches in high molecular weight part to improve the crystallization temperature and increase the viscosity-temperature sensitivity for better preventing the coalesce and collapse during foam-shaping stage.                

4.The cell sizes and expansion ratios of samples should be reported accurately.

The foam expansion ratio was calculated based on the following equation VER=ρ/ρf. Herein ρ and ρf represent the density of polyethylene blends before and after foaming. Density was measured by weighing each sample with a precision of 0.0001 mg and calculating its volume as the volume of water displaced in a calibrated pycnometer where the sample was submerged.                                       

In order to determine cell size (CS) and cell diameter distribution (CSD), a micrograph of each sample was taken using an optical microscope. Cell diameter distribution (CSD) was also obtained by measuring and counting the observed bubbles.

  1. The scale bar of SEM images are not clear.

Sorry, I need to clarify the cells morphology of foamed parts are taken using an optical microscope, not SEM photos. The scale bar are 2000 μm. I provided the original pictures from PE-B1 to PE-B6 for your reference.

Reviewer 2 Report

The paper presents original and interesting research, nonetheless the Materials and Methods section should be further developed. Traceability of work done is of most importance and the information provided is not enough.

Please recheck the paper. Many spaces between words are missing and well as proper typing symbols are not used (example: micrometer symbol).

See attached file.

Author Response

thanks for your good suggestions. please see the attached revised revision. 

Round 2

Reviewer 1 Report

-

Author Response

Dear reviewer,

Thanks for your good suggestions. The revised manuscript has been uploaded the system for your further review.

Reviewer 2 Report

Thank you for addressing my comments. Please check small formatting issues.

Author Response

(The authors gave the same response as above.)
